



# Historical glacier outlines from digitized topographic maps of the Swiss Alps

Daphné Freudiger[1], David Mennekes[1], Jan Seibert[2], Markus Weiler[1]

[1]Chair of Hydrology, University of Freiburg, Freiburg, 79098, Germany
[2]Hydrology and Climate Unit, Department of Geography, university of Zurich, Zurich, 8057, Switzerland

*Correspondence to*: Daphné Freudiger (daphne.freudiger@hydrology.uni-freiburg.de)

**Abstract.** Since the end of the Little Ice Age around 1850, the total glacier area of the Central European Alps has considerably decreased. In order to understand the changes in glacier coverage at various scales and to model past and future streamflow accurately, long-term and large-scale datasets of glacier outlines are needed. To fill the gap between the
morphologically reconstructed glacier outlines from the moraine extent corresponding to the time period around 1850 and the first complete dataset of glacier areas in the Swiss Alps from aerial photographs in 1973, glacier area from 80 sheets of a historical topographic map, so-called Siegfried map, were manually digitized for the publication years 1878-1918 (further called first period, with most sheets being published around 1900) and 1917-1944 (further called second period, with most sheets being published around 1935). The accuracy of the digitized glacier areas was then assessed through a two-step
validation process: the data was (1) visually and (2) quantitatively compared to glacier area datasets of the years 1850, 1973, 2003, and 2010, which were derived from different sources. The validation showed that at least 70 % of the digitized glaciers were comparable to the outlines from the other datasets and were therefore plausible. Furthermore, the accuracy of the manual digitization was found to be lower than 5 %. The presented datasets of glacier outlines for the first and second periods were found to be valuable source of information for long-term glacier mass balance or hydrological modelling in
glacierized basins if the uncertainty of the historical topographic maps is considered in the interpretation of the results. The datasets can be downloaded from the FreiDok plus data repository (https://freidok.uni-freiburg.de/data/12874, DOI: 10.6094/UNIFR/12874).

## 1 Introduction

Total glacier area of the Central European Alps has considerably decreased during the last decades with differences of
change in certain sub-periods (e.g. Fischer et al. 2014). Long-term glacier datasets are of great importance for understanding and assessing glacier changes (Huss & Fischer 2016; Fischer et al. 2015) as well as for hydrological modelling of past and future streamflow (Huss 2011; Viviroli et al. 2011; Stahl et al. 2016). Some glaciers of the Central European Alps have been regularly monitored nearly since the end of the Little Ice Age (ca. 1850), but the majority was only recently or sporadically monitored and long time series of glacier data, except some length changes, are rarely available (WGMS, 2015; GLAMOS,
2015). Remote sensing offers unique opportunities to derive glacier outlines, areas and glacier mass balance at the large scale and several manual and (semi-) automated algorithms were developed in the last decades to identify the entire glacier area of





the Central European Alps up to 1973 (e.g. Maisch et al. 2000; Paul et al. 2011; Fischer et al. 2014; Kääb et al. 2002). Assuming that the end of the Little Ice Age represent the largest glacier extent (Vincent et al. 2005; Collins 2008; Ivy-Ochs et al. 2009), it was even possible to estimate the glacier area around 1850 based on the location of the moraines from this recent maximum glacier extension. The moraines were mapped based on historical topographic maps, field observations and

aerial photographs from the years 1973, 1988, and 1989 (Müller et al. 1976; Maisch et al. 2000; Maisch et al. 2004; Maisch 1992). However, for the time period between 1850 and 1973 no information can be obtained from satellite images analysis and other data sources like aerial photography are only available locally.

The first topographic surveys started in 1809 in Switzerland leading between 1845 and 1864 to the publication of the first

topographic maps for entire Switzerland (including the Alps) based on geometric measurements at a scale of 1:100'000, the Dufour map. During the second half of the 19[th] century, several improvements of the cartographic and drawing methods took place. For example, triangulation with angles was introduced (ca. 1870), the elevation of the "Pierre du Niton" was measured (1879) to be used as reference and the depth of the major Swiss lakes was assessed for the first time (ca. 1870). With these improvements,  new possibilities were given to improve mapping and especially the representation of glaciers in remote

regions (Imhof 1927). The Siegfried map was then produced between 1868 and 1949 following two Federal Acts from 1868 and was based on the Dufour map. The aim was to create for entire Switzerland, especially for the Alps, a homogeneous map system for the Topographic Atlas of Switzerland at a scale of 1:50'000 for the Alps and 1:25'000 for the rest of Switzerland. The project started under the direction of the Chief of Staff, Hermann Siegfried, but most of the mapping was done by cartographers and topographers from the private sector. To insure homogeneity, precise instructions were set from the

beginning (Imhof 1927; Swisstopo). The sheets of the Siegfried map are either simply revision of the Dufour map or were completely newly created. The Siegfried map was considered at the time it was published as the most advanced topographic map ever produced; especially impressive was the drawing in the mountainous regions and the representation of rocks e.g. in glacierized areas (Imhof 1927). Such historical topographic maps provide unique information sources on large scale glacier areas for the time period 1868-1949. They are, however, linked to uncertainties due to the mapping methods available at the

time and possible errors in geo-referencing. Such uncertainties may sometimes lead to inaccuracies when glacier areas from historical maps are compared to other products, for example glacier areas from remotely sensed data (Imhof 1927; Hall et al. 2003; Racoviteanu et al. 2009).

The aim of our study was (1) to manually digitize the historical Siegfried map at two time slices between 1892 to 1944; (2)

to validate the digitized glacier areas through their comparison with glacier areas of different time period and from different data sources in order to assess their accuracy; and (3) finally to create a dataset suitable to be used for example in long-term studies of glacier changes or hydrological modelling.



## 2 Data

### 2.1 Description of the Siegfried map of the Swiss Alps

The Siegfried map covers entire Switzerland and consists in total of ca. 550 sheets that were revised at different publication years. Each sheet covers an area of 210 km$^2$ at a scale of 1:50'000 (alpine regions) or 52.5 km$^2$ at a scale of 1:25'000 and

5 elevation lines are represented every 30 m and 10 m respectively (Fig. 1). The glacierized part of the Swiss Alps is covered by 80 sheets that we digitized for the publication years 1878-1918 with highest frequency around 1900 (further called first period) and 1917-1944 with highest frequency around 1935 (further called second period) as shown in Fig. 2. The Siegfried map was made available by the Swiss Federal Office of Topography (Swisstopo), detailed information on the product can be found on the Swisstopo homepage.

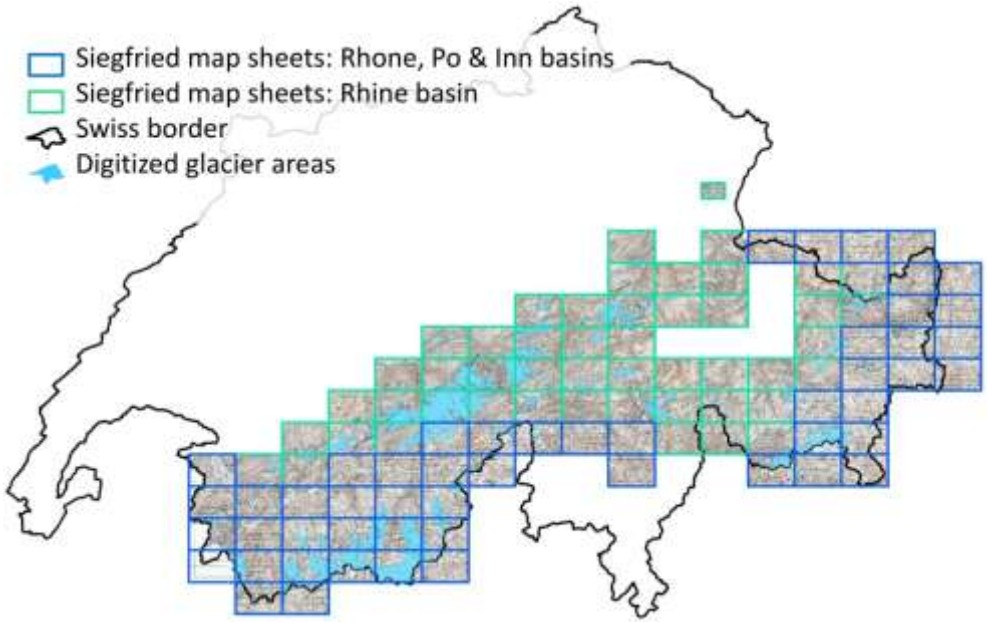

Figure 1: Siegfried maps covering the alpine region of Switzerland.

The arithmetic precision requested from the topographers for the creation of the Siegfried map was 0.7 mm in the projection

15 on the map (corresponding to 35 m in nature) in the alpine region (1:50'000). Error in the representation of the elevation lines can be found, as the reference point of the "Pierre du Niton" was at the time of the map creation erroneously measured 3.26 m above its actual elevation (Imhof 1927). Error ranging from 3.26 to 18 m can therefore be found in the given elevation lines in the maps (Imhof 1927). Furthermore the measurements directives changed during the creation of the maps. At the beginning (around 1880) 300-500 measurements points were needed for the creation of one sheet, while at the end of





the 19<sup>th</sup> century, up to 6000 measurement points were prescribed (Imhof 1927). Large regional differences exist therefore in the horizontal accuracy of the different sheets that is difficult to exactly estimate (Hall et al. 2003; Rastner et al. 2016).

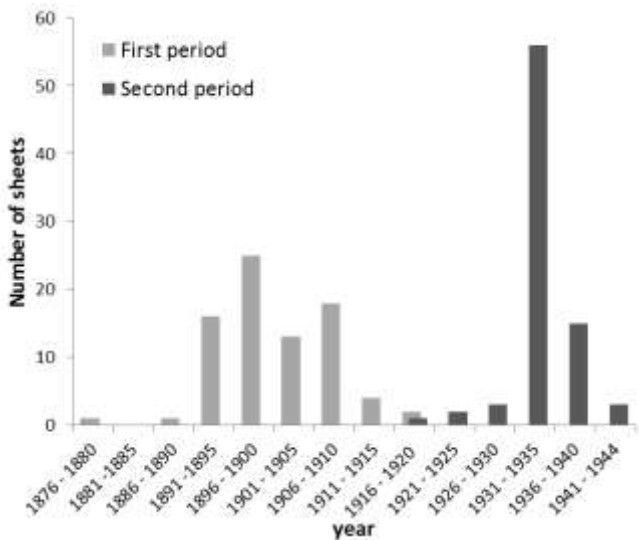

5 **Figure 2: Frequency distribution of the publication years of the 80 sheets from the Siegfried map for the first and second periods.**

## 2.2. Glacier areas and outlines for validation

Four datasets of glacier areas and outlines covering the Swiss Alps for the years 1850, 1973, 2003, and 2010 (Müller et al. 1976; Maisch et al. 2000; Paul et al. 2011; Fischer et al. 2014) were made available for the validation of the digitized glacier 10 areas of the Siegfried map for the first and second period (around 1900 and around 1935). All datasets used for the validation were produced independently with different technologies and methodologies summarized in Table 1.



**Table 1:** Glacier area datasets used for validation.

| Year | Description | Format | References |
|------|-------------|--------|------------|
| 2010 | Aerial ortho-imageries acquired between 2008 and 2011. | shp | Fischer et al., 2014 |
| 2003 | Landsat TM scenes acquired in autumn 2003 | shp | Paul et al., 2011 |
| 1973 | Aerial photographs from September 1973 | gif / tif | Müller et al., 1976; Maisch et al., 2000 |
| 1850 | Glacier outlines morphologically reconstructed from moraine extents of retreated glaciers from aerial photographs of 1973, 1988, and 1989, historical topographic maps, and field observation. | gif / tif | Müller et al., 1976; Maisch, 1992; Maisch et al., 2000; Maisch et al., 2004 |

## 3 Digitization of the Siegfried map

### 3.1 Digitization

All glacier areas of the 80 sheets from the Siegfried map were manually digitized using ArcMap 10.2.2 to create two shape-files with the digitized glacier areas of the first ($A_{S, first}$, around 1900) and second ($A_{S, second}$, around 1935) periods. Special care was taken to exclude outcrops within the glaciers from the digitized glacier areas. For the digitization, the study area was divided into two regions, the Rhine basin and the Rhone, Po, and Inn basins that were digitized by two different persons (Fig. 1). For homogeneity purposes, all digitized areas were finally visually controlled by a single and third person. Altogether, more than 500'000 nodes and 250 working hours were needed to create the polygons and resulting shape files.

### 3.2 Data validation

To assess the quality and accuracy of the digitized glacier areas, $A_{S, first}$ and $A_{S, second}$ were compared with the glacier outlines of the four products available ($A_{1850}$, $A_{1973}$, $A_{2003}$, and $A_{2010}$) in a two-step validation process. As no other contemporary datasets were available to validate the digitized areas, it is not possible to give the exact accuracy of the glacier areas from the sheets of the Siegfried map. However, the two-step validation process presented below allowed us to assess if the digitized glacier areas were consistent with the other products, meaning that the glacier area given in $A_{S, first}$ and $A_{S, second}$ followed a logical evolution compared with the other products.

In a first step, the shapes of the digitized glaciers $A_{S,first}$ and $A_{S,second}$ were visually compared to the other glacier products in order to ensure that they were consistent compared to all datasets. During this comparison, the digitized glacier outlines from the first and second digitized periods that appeared in none of the other products were removed as the existence of a glacier



in this location could not be verified. This was the case for 61.6 km$^2$ and 49.8 km$^2$ of the digitized area of $A_{S,\,first}$ and $A_{S,\,second}$, respectively.

To allow comparison between the different glacier areas of the different data sources and available years, $A_{S,\,first}$, $A_{S,\,second}$, $A_{1850}$, $A_{1973}$, $A_{2003}$, and $A_{2010}$ were divided into 957 glacier areas with an unique identity number, further referred as glaciers, based on the river basin delineations given by the Federal Office for the Environment (FOEN) and following the recommendation of the GLIMS Analysis Tutorial (Racoviteanu et al. 2009). The basins were chosen in the best possible way to cover each individual glacier. However, this delimitation was only used for the aim of comparison and do not represent the real delineation of a glacier area.

Assuming that all glaciers reached their maximum extent at the end of the Little Ice Age around 1850 in the Central European Alps (Vincent et al. 2005; Collins 2008; Ivy-Ochs et al. 2009), the glacier areas from 1850 should be the largest possible. In a second validation step, $A_{S,\,first}$ and $A_{S,\,second}$ were therefore compared to $A_{1850}$, a product derived from the extent of the moraines identified from aerial photographs (Müller et al. 1976; Maisch et al. 2000). We then set following conditions to assess the accuracy of $A_{S,\,first}$ and $A_{S,\,second}$:

- Highly consistent: $A_S < A_{1850}$
- Consistent: $(A_S - A_{1850})/ A_{1850} < 0.1$
- Poorly consistent: $0.1 > (A_S - A_{1850})/ A_{1850} < 0.5$
- Not consistent: $(A_S - A_{1850})/ A_{1850} > 0.5$

As land cover classification in remotely sensed data is not unequivocal (e.g. Racoviteanu et al. 2009) and definition and recognition of moraine partly relies on interpretation (Clark et al. 2004), $A_{1850}$ also shows uncertainties and we therefore considered $A_S$ as consistent with $A_{1850}$ if $A_S$ was up to 10% larger than $A_{1850}$. In case of 'poor consistency' or 'no consistency' between the datasets, $A_S$ and $A_{1850}$ were further compared to the glacier areas of the further available products ($A_{1973}$, $A_{2003}$, and $A_{2010}$) to decide, which one of the two products was more plausible. For this comparison, the shape of each digitized glacier area with poor or no consistency (in total 314 glaciers) was visually compared to the shape of $A_{1973}$, $A_{2003}$, and $A_{2010}$. It was assumed that glacier area decreased between 1850 and 2010 and that $A_S$ or $A_{1850}$ was more likely to be exact if its shape was most corresponding and overlapping the shape of the more recent years ($A_{1973}$, $A_{2003}$, and $A_{2010}$). This evaluation process was entirely realised by one single person for homogeneity reasons and allowed to further assess the accuracy of the digitized maps. From this comparison came two further categories:

- $A_S$ more consistent than $A_{1850}$: When the shape and area of the digitized maps was more in agreement with $A_{1973}$, $A_{2003}$, and $A_{2010}$ than $A_{1850}$; and



- Not consistent but plausible: When it could not be decided from the glacier shape of $A_S$ or $A_{1850}$ which one was more plausible. In this case, both datasets provided plausible glacier shapes but their areas were not comparable.

**Table 2:** Validation of the digitized glacier outlines showed, as an example, for $A_{s, first}$. The third and sixth column represent the sum for the categories 'highly consistent' - 'consistent' - 'more consistent', 'not consistent but plausible', and 'poorly consistent' - 'not consistent'.

| | Number of glaciers | % | Sum (%) | Area (km$^2$) | % | Sum (%) |
|---|---|---|---|---|---|---|
| **Highly consistent** | 478 | 49.95 | | 887.07 | 50.09 | |
| **Consistent** | 165 | 17.24 | 71.16 | 660.54 | 37.30 | 88.34 |
| **More consistent than A$_{1850}$** | 38 | 3.97 | | 16.65 | 0.94 | |
| **Not consistent with A$_{1850}$, but plausible** | 123 | 12.85 | 12.85 | 121.83 | 6.88 | 6.88 |
| **Poorly consistent** | 113 | 11.81 | 15.99 | 68.16 | 3.85 | 4.78 |
| **Not consistent** | 40 | 4.18 | | 16.55 | 0.93 | |
| **Total** | 957 | | | 1770.80 | | |

The results of the validation are presented in Table 2 and Fig. 3 as an example for $A_{s, first}$. Overall 71% of the digitized glaciers of $A_{S, first}$, and even 88% in terms of glacier area were consistent compared with the datasets and for 13% of the glaciers it was not possible to assess if $A_{S,first}$ or $A_{1850}$ was more plausible (Table 2). The results for $A_{s, second}$ are similar with 70% of the glaciers and 89% of the total glacier area being consistent with the other products. The difference between the percentage of digitized glaciers (>70%) and glacier area (>88%) indicates that small glaciers have the highest probability to be inaccurate, which can also be observed in the spatial representation of the validation (Fig.3, as an example for $A_{s, first}$). The total glacier areas for the first period (around 1900) was 1771 km$^2$ and 1711 km$^2$ for the second period (around 1935),while the total reconstructed glacierized area of 1850 was ca. 1735 km$^2$ (Maisch et al. 2000). The difference between the total glacier area around 1900 and 1850 can partly be explained by the fact that certain glaciers were not represented in the 1850-dataset but exist in the other datasets (1973, 2003, and 2010) or were considered as inconsistent compared to the other products (in total: ca. 17 km$^2$).



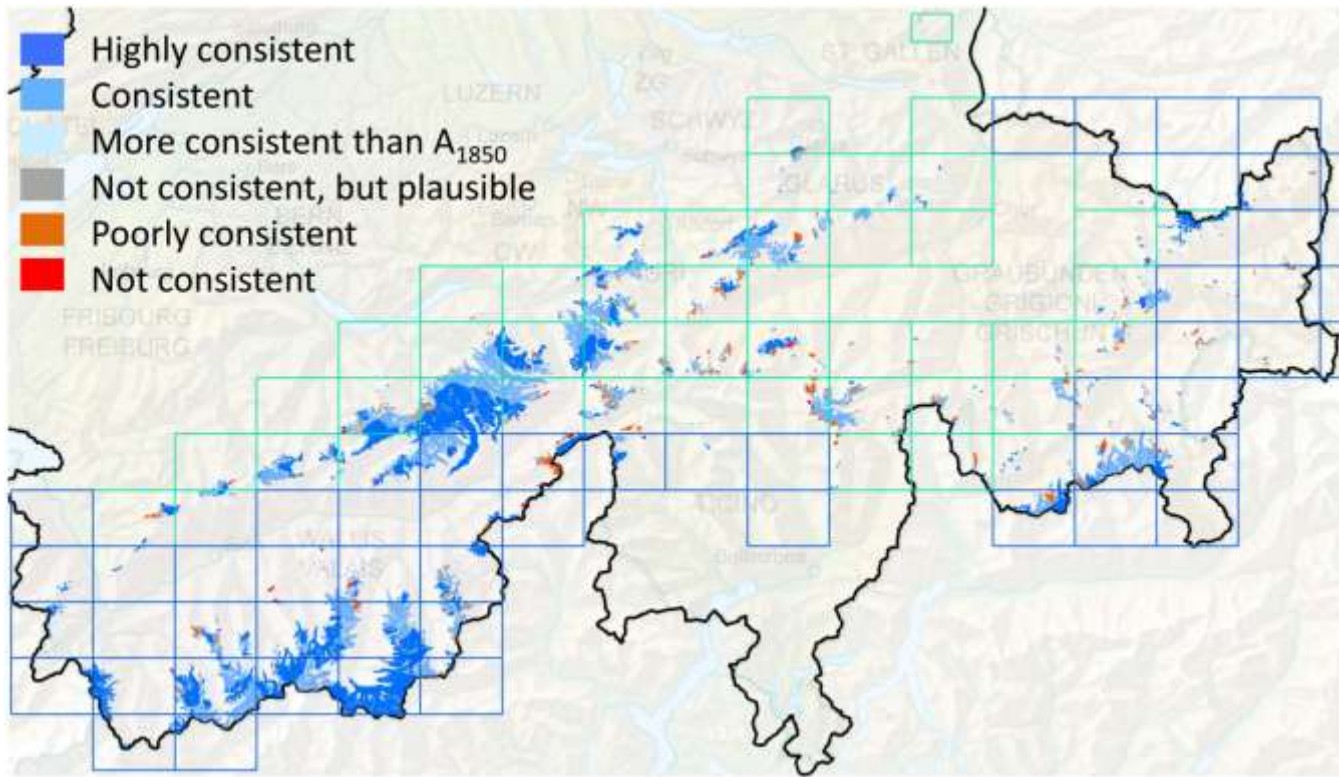

**Figure 3: Validation of the digitized glacier outlines $A_{S,first}$ (around 1900).**

5   **3.3 Accuracy of the digitization**

Similar to mapping glacier areas from remotely sensed data, errors can be induced in the digitizing process of historical topographic maps due to the precision and the interpretation of the map. To assess the accuracy of the digitization, five students (three women and two men between 20-25 years old) were asked to digitize all glaciers over a 23 km$^2$ area from a sheet of the Siegfried map for the publication years 1894 (within the first period) and 1934 (within the second period). The

10   relative error, defined as the variability among the students, increases with decreasing glacier area when glacier outlines are derived from remotely sensed data (e.g. Paul et al. 2013). Therefore, the region of the Wyttenwasser glacier was chosen, which contains rather medium to small glaciers. In Fig. 4 the digitized area of the Wyttenwasser glacier is shown for the publication year 1934 and several sources of conflict for the digitization are pointed out (cases A-C). In case A, the glacier outline in the map is clearly drawn and the differences in the digitized outlines are small and depend only on the precision

15   level of the students. In case B, larger differences are observed, as the map drawing had to be interpreted, e.g. where do the glacier area stops – at the blue topographic line or at the limit between the white area and the black dots? Are the black dots on the glacier area covering ice or is it rock? Where does the glacier tongue ends between a blue and a black topographic line? In case C, one part of the glacier area is overprinted with text, which leads to different interpretation on the glacier



Earth System
Science
Data



outline behind the text. Cases B and C illustrate well the different assumptions that need to be made during digitization of historical topographic maps leading to uncertainties. The comparison of all digitized glacier areas for the year 1894 and 1934 resulted in differences of up to 5% between the five students. These results are comparable to the differences in standard deviation of 2 to 18% for very small glaciers ($< 1km^2$) and smaller than 5% for larger glaciers ($>1km^2$) observed by Fischer et al. (2014) and Paul et al. (2013) while deriving glacier outlines from remotely sensed data.

Case A:
• Clear glacier outlines.
• Only small differences in digitalization.
• Outcrops digitized only by one student.

Case B:
• No clear glacier outlines.
• Great differences in digitalization due to different interpretations.

Case C:
• Clear glacier outlines. Presence of text.
• Great differences in digitalization due to:
(1) different levels of precision; and
(2) assumption of what is behind the text.

**Figure 4: Examples of conflicts encountered during digitization of historical maps. The map shows the glacier area digitized for the end product (blue area with black outlines) and in the background the sheet of the Siegfried map for the publication year 1934 for the Wyttenwasser glacier. In cases A to C the outlines of the same glacier area digitized by five students is shown (coloured lines).**



### 3.4 Accuracy of the digitized glacier areas

While the uncertainty of the digitization could be estimated at around 5%, it is difficult to estimate the accuracy of the Siegfried map itself. As the uncertainty is different for each sheet (see Section 2.1), large regional differences can be found in the accuracy of the glacier outlines and on some sheets, the inaccuracy of the Siegfried map might be much higher than

the interpretation bias of the digitization. However, the two-step validation process allowed us to assess which ones of the digitized glacier areas followed a logical evolution in shape and area and were therefore plausible compared with the other available products of glacier outlines for different years. 71% of the glaciers and 88% of the glacier area were considered as consistent through the analysis. While the presented product of glacier outlines contains all digitized glacier areas from the Siegfried map for Switzerland (Section 4), we recommend to only use the glacier areas that were stated as 'consistent',

'highly consistent', or 'more consistent than $A_{1850}$' by the two-step validation process. If the other glacier areas need to be used, their large uncertainty should be considered in the interpretation of the results.

### 4 Data availability

The datasets of glacier area for the first and second digitized period (around 1900 and around 1935) presented in this paper

are freely available from the FreiDok plus data repository (https://freidok.uni-freiburg.de/data/12874) and has the DOI: 10.6094/UNIFR/12874. For both digitized periods, two shape files are available. The first shape file contains the glacier areas delineated from the digitized sheets themselves with the name of the sheet as identification and the year of publication. The sheets that are exactly identical for both periods are identified in a comment field in the shape file (in total 28 from 80 sheets). The second shape file contains the digitized glaciers delineated as described above for the first digitized period (in

total 957) and the second digitized period (in total 948) with unique identification number. As some glacier extents overlap several sheets and might therefore contain several publication years, the information of both shape files cannot be resumed in a single file. For each digitized glacier, the results of the validation are given in the shape file to enable the use of the different categories (see Table 2) depending on the need of the study (see Sect. 3.2). The basin outlines used for glacier delineation are also available in a separate shape file. All shape files were produced in the CH1903_LV03 projection system.

A readme pdf-file provides a detailed data description.

### 5 Conclusions

The historical topographic Siegfried map represents an important source of information and allowed us to digitize glacier outlines for the entire Swiss Alps for two periods around 1900 and 1935, where modern monitoring technologies cannot be

used. One important challenge when using digitized glacier areas from historical topographic maps is to validate the product to ensure the accuracy and plausibility of the dataset. The Siegfried map is namely linked to several uncertainty sources due to e.g. the limitation of the methodologies used for cartography at the publication time and the possible error in geo-referencing. Furthermore, the sheets of the Siegfried map were not always newly created but sometimes only revised and it is not always possible to ensure the actual date of the glacier mapping. However, the digitized glacier areas were validated with

five other datasets from different sources and years and the data analysis showed that at least 70 % of the digitized glaciers and 88% of the total glacier area were comparable for both digitized periods to the glacier areas and shape of the other products and therefore plausible. The uncertainty of the digitization itself was assessed separately and was lower than 5 % which is comparable to the accuracy of deriving glacier outlines and areas from remotely sensed data. The presented datasets for a first period around 1900 and a second period around 1935 are therefore valuable information to fill the gap between the reconstruction of the glacier areas at around 1850 from the moraine extent and the first complete dataset of glacier areas in the Swiss Alps from aerial photographs in 1973. Under consideration of the data uncertainty, the use of the digitized datasets in combination with other existing datasets from remotely sensed glacier areas can provide important information about changes in glacier areas for the last 120 years, which is essential for long-term and accurate glacier mass balance or hydrological modelling in glacierized basins.

## 6 Author contributions

D. Freudiger homogenized, and validated the presented datasets and prepared the manuscript in contribution from all co-authors.

## 7 Competing interests

The authors declare that they have no conflict of interest.

## 9 Acknowledgements

The authors would like to thank Swisstopo for providing the historical topographic Siegfried map and Damaris De for the digitization of part of the glacier areas. We also thank Mirko Mälicke and his students Ruben Beck, Daniela Boru, Helena Böddecker, Verena Lang, Lukas Maier, and Miranda Perrone for assessing the accuracy of the digitization. The digitization of the glacier areas was made within the ASG-Rhein project (Snow and glacier melt components of the streamflow of the River Rhine and its tributaries considering the influence of climate change) funded by the International Commission for the Hydrology of the Rhine Basin (CHR). The first author was funded by the German Federal Environmental Foundation (DBU).

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
