# Peer review of "Historical glacier outlines from digitized topographic maps of the Swiss Alps"

_Earth System Science Data, 2017_

## Referee Comment (RC1) · Anonymous Referee #1 · 25 Sep 2017

The authors are presenting a dataset of the historical glacier extend in the Swiss alps during the period between 1917 and 1944. The dataset was derived by digitizing the famous Siegfried maps, the most accurate cartographical representation of Switzerland at that time and a worldwide milestone in the history of cartography and mapping of alpine regions. The dataset fills an important data gap between the Swiss Glacier Inventories of 1850 (Maisch, 2000) and 1973 (Müller, 1976). The paper describes a potentially important contribution to the Swiss Glacier Inventories (SGI). The choice of the Siegfried maps is obvious and the quality of the maps allows the assembly of a dataset of the glacier cover during the period of the land survey and map making. Although the compilation of this dataset is an important contribution, I have some substantial comments to be addressed by major revisions before the paper can be accepted. Substantive comments: - Accuracy of glacier outlines: The overall accuracy of the map content is directly related to the density of individual survey stations per map sheet. As described, the number of stations increased from 300 to 6000 per map sheet over the period of the Siegfried map production. The accuracy of 35m in nature (0.7mm in map projection) only describes the accuracy of clearly defined features and survey stations and not topographically interpreted features (Eidgenössische topographische Büreau, 1872) (e.g. rivers, lakes, forest, glaciers, . . .). The accuracy of those features are only depending on the density of surveyed stations (Caminada, 2003; p. 111). The only possible accuracy assessment of the glacier outlines is a cross-validation of the original field survey sheets. For a representative number of glaciers and areas a comparison with the original field survey sheets has to be conducted. The authors have to show, how the density of survey marks are influencing and improving the accuracy of the outlines. An assessment of the same region with different survey dates has to be done. As shown in Caminada (2003) such kind of assessment will allow a proper cross-comparison of glacier outlines.

- Dating of glacier outlines: As main source for the determination of the date of the glacier outlines the year of the map publication is used. Due to the very long process of land survey and map production this assumption is misleading and not accurate. At the best, the year of publication can be used as an indicator for further assessments. An accurate determination of the dates will be only possible by using and cross-comparison of the original field survey sheets. An important contribution is Bauder et al. (2007) with the comparison of recent datasets and datasets derived from the Siegfried maps.

- Subdivision of glacier outlines: As subdivision of the glacier outlines, catchment geometries were used (river basin delineations). This subdivision will be useful for hydrological assessments and models, but will not be practical for glaciological studies. Therefor the title of the paper is misleading and should contain a detailed definition of the possible usage of the dataset (e.g. hydrological modelling). The digitizing and

the subdivision is not taking in account glaciological principals of glacier subdivisions. The naming of the glacier geometries is neither following the GLIMS identification nor the Swiss Glacier Inventory nomenclature. Therefore, the usage of the dataset for the comparison with other releases of the SGI is not recommended. Comparisons of the dataset is only possible by spatial intersection which is due to the inaccuracy of the geometry misleading. A valuable comparison of the dataset will be only possible by using a logical approach based on common identifiers of each glacier (e.g. GLIMS ID, SGI ID).

- Statistics of numbers of glaciers: The term "Number of glaciers" is misleading and due to the methodology of subdivision (catchments) not accurate. Currently the dataset can only be quantified by the surface (km2) or by the total length of outlines digitized (km).

- Validation of glacier outlines: The method of the validation is straight forward and could be applied as a first assessment. It is obvious that several regions (e.g. South-East of the Bernese Alps) are mainly classified as "Consistent". The zoning of "Highly consistent" and "Consistent" follows major watersheds or cantonal boundaries. This zoning looks systematically and has to be revised. Base on the current approach neither the geometrical nor the temporal precision is known. The validation in the current paper is only a qualitative comparison between two datasets. A proper validation can only be done with a temporal comparable source (see below).

- Validation of glacier outlines with additional sources: A proper validation of the glacier outlines of the Siegfried maps should be done with an additional dataset at more or less the same period of time. Such kind of source can be the first edition of the topographical maps in a scale of 1:50'000 which followed the Siegfried maps. Based on the assessment of the survey dates (Mercanton, 1958) a more detailed dating of the outline can be done.

Detailed comments:

- Page 2, Line 25: The determination, localisation and handling of spatial and temporal

uncertainties should be the key content of the article. An in-depth assessment of these uncertainties will be the most important contribution for the further usage of the dataset. The paper and the dataset has to include a suitable methodology and attribution of uncertainties. This has to be done with available original sources (survey sheets and notes) and cross-comparions of further datasets of the same period of time.

- Page 3, Line 14: The arithmetic precision is only valid for survey stations and clearly defined landmarks. But not for glacier outlines.

- Page 3 Line 17: Correct. But the error is gradually from West to East and could be even roughly determined. For this paper, the vertical accuracy is not relevant. The statement can be omitted.

- Page 4, Line 1: The large differences of the density of measurement points is the main factor of the accuracy. This has to be reviewed and quantified.

- Page 4, Line 2: Raster et al. (2016) is providing an extensive analysis and methodology of accuracy assessment using Siegfried maps.

- Page 4, Line 11: The datasets of 1850, 1973 and 2010 are not produced independently. On the contrary they are highly depending due to the fact, that they were used as references.

- Page 5, Line 11: The total number of nodes and working hours are not relevant. Relevant is the relationship between total numbers of points compared with the total length digitized and the reference scale used during digitizing.

- Page 6, Line 9: Correct. The current delineation is not taking in account glaciological principals. Therefore, the authors should use the misleading term "Number of glaciers".

- Page 8, Line 8: Sex and age of the persons are not relevant. Relevant are the skills of map reading and interpretation of glaciological phenomena.

- Page 8, Line 11: Main factor for the comparison of different datasets digitized are

the usage of common guidelines (e.g. scale, interpretation, . . .). Otherwise the competence of the operators is will be compared and not the precision of the digitizing process.

- Page 8, Line 14: In this approach, the capacity of the operators is compared and not the process itself.

- Page 10, Line 4: The inaccuracy of the Siegfried map is higher than the process of digitizing. Instead of analysing the inaccuracy of the digitizing process, the accuracy of the individual map sheets has to be investigated.

- Page 10, Line 17: The year of publication is irrelevant and misleading. An accurate assessment of the original data has to be done.

- Page 10, Line 34: With the current methodology, the date of the glacier mapping is never accurate. The year of map edition is by far not a reliable source of information.

Caminada, P. (2003). Pioniere der Alpentopografie : die Geschichte der Schweizer Kartenkunst. Zürich : AS-Verlag.

Eidgenössische topographische Büreau (1872). Instruktion für topographische Aufnahmen im Massstab 1:50000.

Mercanton, P. L. (1958). Aires englacées et cotes frontales des glaciers suisses, leurs changements de 1876 à 1934 d'après l'Atlas Siegfried et la Carte Nationale et quelques indications sur les variations de 1934 à 1957. Zurich.

Bauder, A., Funk, M., & Huss, M. (2007). Ice-volume changes of selected glaciers in the Swiss Alps since the end of the 19th century. Annals of Glaciology, 46, 145-149. doi:10.3189/172756407782871701

---

## Referee Comment (RC2) · Anonymous Referee #2 · 26 Oct 2017

General comments: I found this to be a highly interesting methodological paper at the top of its field, developing new methodologies for analysing longer-term glacier change. Although methods for analysing glacier change from satellite images are well established, there are few published protocols for digitising topographic maps. However, these topographic maps offer the opportunity to extend the glaciological record far beyond the satellite era. Methodological papers such as this are therefore highly welcome. This paper is well written and clear.

The evaluation and validation of the digitization of the topographic maps is especially nice work. There is little analysis of the glacier recession trends and dynamic; I assume therefore that this is coming in a companion paper. If this is not the case, then the paper should include further analyses of the trends in glacier changes and hydrological

modelling.

Specific comments:

I have few comments regarding the paper. In places the phrasing is slightly awkward or unclear and could be tightened.

In section 3, it was unclear to me whether the glacier outlines were mapped in this paper, or whether glacier outlines were mapped by previous authors and imported into this work. If they authors did not map the glacier outlines from A1850, A1973 and A2010, how comparable are they as surely different methods were used? If the authors mapped the glacier outlines themselves, then further detailed description and evaluation of this is required; comparing glacier outlines derived from multiple different methods and by different researchers should be discussed.

For the map survey sheets, when did the surveying take place? Did it use orthorectified aerial photographs and when were these taken? How different is this to the date of map publication? The delimitation of the glaciers into different catchments – why not use the same ice divides and catchments as the previously published GLIMS glaciers?

Technical comments:

Line 17: "inaccuracy"?

Line 28: "majority were only recently. . ."

Line 2: "Little Ice Age represents the largest. . ."

Awkward phrasing in this paragraph, unclear.

Line 10: Do these maps have a reference and publication information?

Line 13: "(2879 m asl)"

[Figure]

Page 8 – I would question whether the ages and genders of the students is necessary information. Are they undergraduate students? Or graduate students? This may be pertinent. How were they selected? Was this part of an undergraduate research project? How reliable and robust are their results? Their degree of training has implications for their skill level in interpreting the topographic maps and mapping the glaciers.

---

## Author Comment (AC1) · 15 Dec 2017

Dear Editor,

We thank the reviewers for their valuable comments, which will help us to improve the manuscript. In particularly, when revising the manuscript, we will:

- Add a cross-comparison of the digitized Siegfried map with contemporary glacier outlines from different sources for some specific glaciers. This analysis will help to better assess the accuracy of the Siegfried map

- Discuss more in detail and clarify (1) the choice of the glacier delineation from river basins and (2) the difference between the publication and survey years

- Add a section about possible application/use of the Siegfried map

Please find below a detailed response to the comments of Referee #1 and Referee #2.

Regards,

Daphné Freudiger on behalf of all co-authors

Response to Referee #1:

******* Referee #1 comment ***** The authors are presenting a dataset of the historical glacier extend in the Swiss alps during the period between 1917 and 1944. The dataset was derived by digitizing the famous Siegfried maps, the most accurate cartographical representation of Switzerland at that time and a worldwide milestone in the history of cartography and mapping of alpine regions. The dataset fills an important data gap between the Swiss Glacier Inventories of 1850 (Maisch, 2000) and 1973 (Müller, 1976). The paper describes a potentially important contribution to the Swiss Glacier Inventories (SGI). The choice of the Siegfried maps is obvious and the quality of the maps allows the assembly of a dataset of the glacier cover during the period of the land survey and map making. Although the compilation of this dataset is an important contribution, I have some substantial comments to be addressed by major revisions before the paper can be accepted. *******************************

We thank Referee #1 for the valuable comments. We address the substantial comments point by point below. The detailed comments will be addressed in the revision process.

Substantive comments:

******* Referee #1 comment ***** - Accuracy of glacier outlines: The overall accuracy of the map content is directly related to the density of individual survey stations per map sheet. As described, the number of stations increased from 300 to 6000 per map sheet over the period of the Siegfried map production. The accuracy of 35m in nature (0.7mm in map projection) only describes the accuracy of clearly defined features and survey

stations and not topographically interpreted features (Eidgenössische topographische Büreau, 1872) (e.g. rivers, lakes, forest, glaciers, . . .). The accuracy of those features are only depending on the density of surveyed stations (Caminada, 2003; p. 111). The only possible accuracy assessment of the glacier outlines is a cross-validation of the original field survey sheets. For a representative number of glaciers and areas a comparison with the original field survey sheets has to be conducted. The authors have to show, how the density of survey marks are influencing and improving the accuracy of the outlines. An assessment of the same region with different survey dates has to be done. As shown in Caminada (2003) such kind of assessment will allow a proper cross-comparison of glacier outlines. *******************************

Response: We agree with Referee #1 that assessing the accuracy of the Siegfried map is of great importance for further use. However, the accuracy estimation of the Siegfried map is limited by the available data and the number of surveyed stations is not easily accessible, which is probably also the reason why Caminada (2003) did not use either this information to assess the accuracy of the Siegfried map. However, we agree that a cross-comparison of the glacier outlines with other contemporary products such as more recent topographic maps and photographs would add information on the accuracy of the Siefried map. This kind of cross-comparison can only be done for glacier outlines where further data is available (not for the entire Alps) but would be a good complement to the large-scale consistency analysis we performed in this study. In the revision process, we will add a cross-comparison with other contemporary products of some specific glaciers.

******* Referee #1 comment ***** -Dating of glacier outlines: As main source for the determination of the date of the glacier outlines the year of the map publication is used. Due to the very long process of land survey and map production this assumption is misleading and not accurate. At the best, the year of publication can be used as an indicator for further assessments. An accurate determination of the dates will be only possible by using and cross-comparison of the original field survey sheets. An

important contribution is Bauder et al. (2007) with the comparison of recent datasets and datasets derived from the Siegfried maps. ******************************

Response: We agree with Referee #1 that the publication year cannot be interpreted as the measurement year. Unfortunately, the publication year is the only information that is available together with the map and the measurement year cannot be given more precisely. In the revision we will therefore add a further comment to make clear to the users, that the year given is the publication year and it should therefore be taken into account that the actual years of field survey was several years before.

******* Referee #1 comment ***** -Subdivision of glacier outlines: As subdivision of the glacier outlines, catchment geometries were used (river basin delineations). This subdivision will be useful for hydrological assessments and models, but will not be practical for glaciological studies. Therefor the title of the paper is misleading and should contain a detailed definition of the possible usage of the dataset (e.g. hydrological modelling). The digitizing and the subdivision is not taking in account glaciological principals of glacier subdivisions. The naming of the glacier geometries is neither following the GLIMS identification nor the Swiss Glacier Inventory nomenclature. Therefore, the usage of the dataset for the comparison with other releases of the SGI is not recommended. Comparisons of the dataset is only possible by spatial intersection which is due to the inaccuracy of the geometry misleading. A valuable comparison of the dataset will be only possible by using a logical approach based on common identifiers of each glacier (e.g. GLIMS ID, SGI ID). Statistics of numbers of glaciers: The term "Number of glaciers" is misleading and due to the methodology of subdivision (catchments) not accurate. Currently the dataset can only be quantified by the surface (km2) or by the total length of outlines digitized (km). ******************************

Response: The subdivision of glacier outlines has been done based on river basin delineations as it was not possible to use the common identifiers for the entire map (SGI ID or GLIMS ID). These identifiers are namely only available for more recent years at the large scale and the changes in glacier areas make a unique identification difficult

(shape or presence of glacier is often different) for earlier years. We therefore decided to follow the recommendation of GLIMS Analysis Tutorial (Racoviteanu et al., 2009) and to delineate the glaciers with basin outlines. As we provide both products, the one delineated on the Siegfried sheets and the one delineated based on river basin, we disagree that the product is limited to hydrological studies, but the proper delineation needs to be done by the user depending on the goal of the study. However, we agree that the term "glaciers" for our delineated glacier outlines might be confusing. We will reformulate this term in the revision.

******* Referee #1 comment ***** - Validation of glacier outlines: The method of the validation is straight forward and could be applied as a first assessment. It is obvious that several regions (e.g. SouthEast of the Bernese Alps) are mainly classified as "Consistent". The zoning of "Highly consistent" and "Consistent" follows major watersheds or cantonal boundaries. This zoning looks systematically and has to be revised. Base on the current approach neither the geometrical nor the temporal precision is known. The validation in the current paper is only a qualitative comparison between two datasets. A proper validation can only be done with a temporal comparable source (see below). Validation of glacier outlines with additional sources: A proper validation of the glacier outlines of the Siegfried maps should be done with an additional dataset at more or less the same period of time. Such kind of source can be the first edition of the topographical maps in a scale of 1:50'000 which followed the Siegfried maps. Based on the assessment of the survey dates (Mercanton, 1958) a more detailed dating of the outline can be done. ******************************

Response: For the glacier outline delineation, we used the smallest aggregation of the river basin delineations provided by the Swiss Federal Office for the Environment (FOEN) which are much smaller than the major watersheds and independent from the cantonal boundaries. The "zoning" following the major watersheds or cantonal boundaries suggested by Referee #1 is therefore not systematic but rather a coincidence. To further validate the product, we will provide a comparison of the glacier outlines of

some specific glaciers with contemporary products (see response above). We will also compare the area from the Siegfried map with the area calculated from the national topographic map (ca. 1934) in Mercanton (1958) for several river basins. The first comparison will allow us to validate the geometry of the digitized glaciers for several glaciers and the second comparison will give more information on the accuracy of the glacier area for some river basins.

Response to Referee #2

******* Referee #2 comment ***** General comments: I found this to be a highly interesting methodological paper at the top of its field, developing new methodologies for analysing longer-term glacier change. Although methods for analysing glacier change from satellite images are well established, there are few published protocols for digitising topographic maps. However, these topographic maps offer the opportunity to extend the glaciological record far beyond the satellite era. Methodological papers such as this are therefore highly welcome. This paper is well written and clear. The evaluation and validation of the digitization of the topographic maps is especially nice work. *****************************

We thank Referee#2 for the kind comments. We answer the main comments below, the technical comments will be assessed in the revision process.

******* Referee #2 comment ***** There is little analysis of the glacier recession trends and dynamic; I assume therefore that this is coming in a companion paper. If this is not the case, then the paper should include further analyses of the trends in glacier changes and hydrological modelling. *****************************

Response: As the scope of the journal is to present the data product and no analysis should be included, we did not include any glacier trend analysis in the manuscript. However, a second manuscript is in preparation where the data is used for a trend analysis of the changes in glacier area from 1850 to 2010. This manuscript should be soon submitted to another journal. Furthermore, the presented dataset has already

been used for hydrological modeling of the Rhine River basin (Stahl et al., 2017) and was included to a glacier routine in order to represent the glacier dynamics over 100 years from 1900 (Seibert et al., 2017). We will add a section in the revised manuscript on possible application/use of the dataset.

\*\*\*\*\*\*\* Referee #2 comment \*\*\*\*\* Specific comments: I have few comments regarding the paper. In places the phrasing is slightly awkward or unclear and could be tightened. In section 3, it was unclear to me whether the glacier outlines were mapped in this paper, or whether glacier outlines were mapped by previous authors and imported into this work. If they authors did not map the glacier outlines from A1850, A1973 and A2010, how comparable are they as surely different methods were used? If the authors mapped the glacier outlines themselves, then further detailed description and evaluation of this is required; comparing glacier outlines derived from multiple different methods and by different researchers should be discussed. \*\*\*\*\*\*\*\*\*\*\*\*\*\*\*\*\*\*\*\*\*\*\*\*\*\*\*\*\*\*\*\*

Response: The glacier products for the years 1850, 1973 and 2010 were digitized and made available by different authors as given in the manuscript in Table 1. For the comparison of the different products, we defined the glacier areas to be compared (A1850, AS,first, AS,second, A1973, A2003, and A2010) as the total glacier area delineated by the smallest aggregation of the Swiss river basins given by the Swiss Federal Office of Environment (FOE). With this method, the glacier areas do not represent the exact outlines of a glacier and it can happen that one "glacier" is composed of few small glaciers. As answered to Referee #1, it was unfortunately not possible to use the delineation using the common identifiers (e.g. GLIMS ID, SGI ID), as this delineation is available for more recent years only. Therefore, we used the delineation method recommended by the GLIMS Analysis Tutorial (Racoviteanu-et al., 2009), which allowed us to have comparable glacier areas for each year. As we realize that this methodology was not clear enough in the manuscript, we will clarify it in the revised manuscript and will further discuss the consequences of the delineation on the interpretation of the results. We will also further discuss the chosen methodology based on the comparison of glacier

outlines from multiple different methods.

******* Referee #2 comment ***** For the map survey sheets, when did the surveying take place? Did it use orthorectified aerial photographs and when were these taken? How different is this to the date of map publication? *****************************

Response: The Siegfried map was created with in-situ measurements and based on older topographic maps (Dufour maps) and no aerial photographs were used. The measurement campaigns could last for several years. Unfortunately, the surveying years are not available with the maps but it can be taken into account that it occurred up to several years before the publication years. As this is an important point, we will discuss it in the revised manuscript (see also comment above).

******* Referee #2 comment ***** The delimitation of the glaciers into different catchments – why not use the same ice divides and catchments as the previously published GLIMS glaciers? ***************************** Response: See comments above.

References:

Caminada, P. (2003). Pioniere der Alpentopografie: die Geschichte der Schweizer Kartenkunst. Zürich: AS-Verlag.

Mercanton, P. L. (1958). Aires englacées et cotes frontales des glaciers suisses, leurs changements de 1876 à 1934 d'après l'Atlas Siegfried et la Carte Nationale et quelques indications sur les variations de 1934 à 1957. Zurich.

Racoviteanu, A. E., Paul, F., Raup, B., Khalsa, S. J. S., & Armstrong, R. (2009). Challenges and recommendations in mapping of glacier parameters from space: Results of the 2008 global land ice measurements from space (GLIMS) workshop, Boulder, Colorado, USA. Annals of Glaciology, 50(53), 53–69. https://doi.org/10.3189/172756410790595804

Seibert, J., Vis, M. J. P., Kohn, I., Weiler, M., & Stahl, K. (2017). Technical Note: Representing glacier dynamics in a semi-distributed hydrological model. Hydrology

and Earth System Sciences Discussions, (March), 1–20. https://doi.org/10.5194/hess-2017-158

Stahl, K., Weiler, M., Freudiger, D., Kohn, I., Seibert, J., Vis, M., ... Böhm, M. (2017). The snow and glacier melt components of streamflow of the river Rhine and its tributaries considering the influence of climate change. Final report to the International Commission for the Hydrology of the Rhine (CHR). Retrieved from www.chr-khr.org/en/publications